# Neglected Fractures of the Lateral Humeral Condyle in Children; Which Treatment for Which Condition?

**DOI:** 10.3390/children8010056

**Published:** 2021-01-18

**Authors:** Giovanni Trisolino, Diego Antonioli, Giovanni Gallone, Stefano Stallone, Paola Zarantonello, Piergiuseppe Tanzi, Eleonora Olivotto, Luca Stilli, Giovanni Luigi Di Gennaro, Stefano Stilli

**Affiliations:** 1Unit of Pediatric Orthopaedics and Traumatology, IRCCS Istituto Ortopedico Rizzoli, 40136 Bologna, Italy; diego.antonioli@ior.it (D.A.); giovanni.gallone01@gmail.com (G.G.); stallone.stefano@gmail.com (S.S.); p.zarantonello1@gmail.com (P.Z.); tanzipiergiuseppe@gmail.com (P.T.); luca.stilli@libero.it (L.S.); giovanniluigi.digennaro@ior.it (G.L.D.G.); stefano.stilli@ior.it (S.S.); 2RAMSES Laboratory, RIT Department, IRCCS Istituto Ortopedico Rizzoli, 40136 Bologna, Italy; eleonora.olivotto@ior.it

**Keywords:** neglected fracture, lateral humeral condyle, elbow, delayed union, nonunion, child, in situ fixation, ORIF

## Abstract

Background: Neglected fractures of the lateral humeral condyle (LHC) are misdiagnosed or insufficiently treated fractures, presenting later than 3 weeks after injury. The management of neglected LHC fractures in children remains controversial. Methods: Twenty-seven children were included in this retrospective study. Charts and medical records were investigated for demographics, time interval between injury and treatment, and type of treatment. Baseline radiographs were assessed for fracture grading and displacement. Final radiographs were investigated for bone healing, avascular necrosis, elbow deformities and growth disturbances. Complications were classified by the Clavien–Dindo–Sink (CDS) system. Outcomes were assessed according to the Dhillon Score (DhiS) and Mayo Elbow Performance Score (MEPS). Results: The mean time from injury to presentation was 27 months. Treatments included nonoperative management (6 patients), “in-situ” fixation (7 patients), open reduction and internal fixation (11 patients) and corrective osteotomy (3 patients). The mean follow-up was 7 years (range: 2–16). Overall, we observed complications in 16 patients (59%); six complications were considered major (22%) and occurred in Weiss Grade 3 fractures, with lateral displacement ≥5 mm. At the latest follow-up, pain and functional scores improved in 23 of 27 patients (85%). Mean MEPS increased from an average of 62 points preoperatively to 98 points postoperatively, while mean DhiS improved on average from 5 to 8 points. CDS score and time interval between injury and treatment were independent predictors of MEPS and DhiS. Conclusion: Our study describes outcomes from a cohort of children undergoing different treatments for neglected LHC fractures. Prolonged time interval between injury and treatment and perioperative major complications negatively impacted the treatment outcomes. Our findings strengthen the requirement for widely agreed guidelines of surgical management in neglected LHC fractures.

## 1. Introduction

Fractures of the lateral humeral condyle (LHC) account for 10–20% of pediatric elbow fractures [1]. Although many nondisplaced LHC fractures may be treated conservatively, fractures with >2 mm displacement are considered unstable, thus requiring surgical fixation [2,3,4].

The assessment of LHC fractures may be challenging, especially if the fracture is minimally displaced or occurs in preschool children, when the cartilaginous component is predominant [3]. Neglected LHC fractures are unrecognized or misdiagnosed fractures that present later than 3 weeks after injury [5,6,7,8,9,10]; misdiagnoses may occur in up to 17% of cases, causing inadequate and/or insufficient conservative management and, consequently, malunion, delayed union or nonunion [1]. Moreover, some anatomic and biologic factors, such as impaired blood supply of the metaphyseal fragment, articular fluid interposition among fragments and traction forces by forearm extensors, as well as surgical pitfalls, such as insufficient reduction or stabilization of the fragment, may retard or prevent fracture healing [10,11]. Despite no evidence-based thresholds for fracture healing have been currently established, it is widely accepted that delayed union is a LHC fracture in which no complete bone healing is observed by more than 8 weeks after the initial injury, while nonunion is defined as no progression of bone healing after 3 months [1,5,11,12,13]. Delayed union and nonunion of LHC fractures in children account for 2.5% overall, according to a recent systematic review [1].

The management of neglected LHC fractures remains controversial. Several treatments have been proposed, including nonoperative management [5,6,14], open reduction and internal fixation (ORIF) [7,8,9,10,12,15,16], “in-situ” fixation (ISF) [13,16,17], bone grafting ± ORIF [5], anterior transposition of ulnar nerve (ATUN) and corrective osteotomy (CO) [18,19,20]. To date, the treatment outcomes of neglected LHC fractures in children are underreported (approximately 450 cases described in literature), without evidence-based guidelines concerning their management.

The aim of this study was to describe a cohort of children treated at a single institution for neglected LHC fractures, analyzing complications and identifying potential factors affecting the outcomes, in order to provide specific recommendations for the management of such challenging cases.

## 2. Materials and Methods

### 2.1. Study Design and Patient Selection

Ethics approval was obtained. Parents or caregivers gave consent for study participation, since all the patients were children at the time of inclusion. The administrative database of a tertiary referral hospital for pediatric orthopedics and traumatology was retrospectively searched for children with a diagnosis of neglected LHC fracture between 2002 and 2017. Criteria for eligibility were unstable LHC fracture (displacement > 2 mm) and open physis, presenting later than 3 weeks after injury with no evident signs of fracture healing on plain radiographs. Causes for exclusion were incomplete clinical or radiographic data or follow-up of <2 years.

Treatments were carried out according to the surgeons’ experience and preference. Treatment options included: (1) nonoperative treatment with prolonged cast immobilization (at least 8–12 weeks), continuous or intermittent elbow splint and close radiographic follow-up; (2) ISF through Kocher’s approach, which allowed for debridement of bony ends, bone grafting and stabilization with cannulated lag-screws and/or K-wires, without any attempt at fragment reduction [13,16]; (3) ORIF through Kocher’s approach; (4) medial closing-wedge varus osteotomy and ATUN through a medial approach, without any attempt to repair the nonunion.

Charts and medical records were reviewed by three independent reviewers (S.Sta., P.Z. and P.T.) and investigated for demographics, age at treatment, time interval between injury and treatment, initial treatment of the fracture and subsequent treatment of the delayed union or nonunion. Radiographs were assessed by two orthopedic surgeons (G.T. and D.A.) with more than 10 years’ experience in pediatric orthopedics. Baseline radiographs were assessed for fracture grading (according to Milch, Weiss and Song classifications) and displacement [3,13,21]. The amount of displacement was measured on radiographs of the anteroposterior and internal oblique view. Medial displacement was measured from medial metaphyseal end of the lateral condyle fragment to its original site of distal humerus, and lateral displacement was measured from the lateral metaphyseal end to its original site. The maximal values measured on the anteroposterior or internal oblique view were regarded as the amount of medial and lateral displacement [13]. Final radiographs were investigated for bone healing, avascular necrosis (AVN), elbow deformities and growth disturbances, such as physeal arrest, bone overgrowth or fishtail deformity.

Complications were classified according to the Clavien–Dindo–Sink (CDS) system; those with CDS > 2 were considered major complications [4,22]. Clinical and functional scores were recorded at baseline and at final follow-up. Clinical outcomes were assessed according to Dhillon Score (DhiS) and Mayo Elbow Performance Score (MEPS) [6,23]. Range of motion (ROM) and elbow carrying angle were measured with a goniometer. Clinical and functional outcomes were assessed by an orthopedic surgeon with more than 20 years’ experience in pediatric orthopedics and shoulder and elbow surgery (G.L.D.), supported by two experienced residents (P.T., G.G.) after adequate training.

### 2.2. Statistical Analysis

Continuous variables were expressed as means ± standard deviation (SD) and ranges. Categorical variables were expressed as raw numbers and percentages. Normality was tested using the chi-square test for categorical variables and the Kolmogorov–Smirnov test for continuous variables. Fisher’s exact tests were used for comparisons of categorical variables. Parametric and nonparametric methods (ANOVA, Mann–Whitney and Kruskal–Wallis tests) were used for comparison of continuous variables among groups, depending on the normality of distributions. Bonferroni’s correction was applied for multiple comparisons. Continuous variables within groups were compared using Wilcoxon’s signed-rank test. Relationships between quantitative variables were assessed using Pearson’s or Spearman’s correlation depending on the normality of distributions and a coefficient *r* > 0.4 was considered to be relevant. Exploratory univariable and multivariable analyses were performed to identify potential associations between baseline variables and outcomes. Variables that were significantly associated with the outcomes were included in multiple regression models to identify potential predictors of outcomes. A *p*-value < 0.05 was considered statistically significant. All data were analyzed using SPSS 22.0 (IBM, Armonk, NY, USA).

## 3. Results

### 3.1. Patient Demographics and Baseline Characteristics

Thirty-three patients were identified. Among them, six patients were excluded (two patients because they were further diagnosed as malunion, one patient because of skeletal maturity at presentation, three patients for incomplete documentation), leaving 27 patients for final inclusion. Demographics, baseline characteristics, intervention and outcomes are reported in Table 1 and Table 2. Raw data are reported in the Appendix A.

There were 20 boys and 7 girls (mean age at diagnosis: 6.3 years; range: 1.7–14.7). The right elbow was involved in 18 patients. All patients but three were referred to our institution from other hospitals. LHC fracture was initially managed nonoperatively in 21 children (78%). The reasons for inadequate nonoperative management were missed diagnosis (*n* = 3), underestimation of fracture severity (*n* = 16) and secondary displacement missed at intermediate follow-up visits (*n* = 2). The remaining six patients (22%) had failure of ORIF with K-wires. 

The mean time from injury to presentation was 27 months and was significantly longer in children undergoing CO (Table 1). Twelve children presented within 3 months after injury and thus were classified as delayed union, while 15 children presented at more than 3 months after injury and were considered nonunion.

Presenting symptoms were variable: 23 patients (85%) had pain and/or stiffness; four children (15%) had no symptoms. Sixteen patients (59%) had normal or nearly normal ROM (arc > 100°). Twelve patients had flexion contracture greater than 30°, so the carrying angle could not be correctly assessed. Eight patients presented for varus (*n* = 2) or valgus (*n* = 6) deformity; three children with valgus deformity showed ulnar neuropathy.

On radiographs, the initial fracture was classified as Milch 2 in 25 patients (93%), as Grade 2 (41%) or 3 (59%) according to Weiss, and as Grade 3 (59%), 4 (33%) or 5 (7%) according to Song. Lateral displacement averaged 7 ± 4 mm; medial displacement was 9 ± 4 mm. 

### 3.2. Treatment and Outcomes

Six children underwent nonoperative treatment. Five of them, presenting within 3 months from injury with ≤5 mm displacement, showed successful healing without further displacement at a mean follow-up of 7 months (3.5–13) after injury. At the latest follow-up, these patients had excellent clinical and functional outcomes. Radiographs showed successful bone remodeling with bony prominence in three patients and slight fishtail deformity in one patient (Figure 1). 

The remaining patient presented 14 weeks after injury with severe displacement (Figure 2a). ORIF was recommended but the parents refused treatment, since the child was asymptomatic and they were worried about possible complications. Therefore, this patient was included in the nonoperative group for the purposes of the study. This child presented 5.5 years later with persistent nonunion and severe valgus elbow deformity (Figure 2b–d). CO was performed, without repairing the nonunion, to avoid elbow stiffness (Figure 2e). At the latest follow-up, the child had a good outcome with no pain or impaired function, 5° extension deficit, 10° flexion deficit and a residual valgus of 10°, compared with the contralateral elbow.

Seven children presenting 9 months (range: 2–45) on average after injury, with a mean lateral displacement of 4 mm (range 3–6), underwent ISF. Of these, six patients successfully healed at a mean follow-up of 2 months (1–6). One child undergoing ISF with two K-wires had no progression of healing within 6 months after surgery. This patient had further surgery with screw fixation and bone allograft. The nonunion then healed uneventfully 4 months later, but the child sustained another non-displaced LHC fracture 18 months after surgery, which healed satisfactorily with cast immobilization for 4 weeks.

Eleven children underwent ORIF at a mean time of 24 months (Figure 3). Children undergoing ORIF had a significantly higher fracture grade according to Song (*p* = 0.015) and larger lateral displacement (8 ± 2 mm versus 4 ± 1 mm; *p* < 0.0005). Nonunion healed in all patients within 6 months (1–25). A child developed AVN and another child developed malunion that required further surgery. Moreover, two children had important varus or valgus angulation at baseline that was not addressed during the operation and remained substantially unchanged at the latest follow-up.

Three children presented with long-standing nonunion associated with severe valgus deformity and ulnar neuropathy. Mean age at presentation, time interval between injury and treatment, and baseline carrying angle were significantly higher in this group compared with the other groups. At the latest follow-up, one girl had valgus recurrence with persistent ulnar neuropathy, while the other two boys had good outcomes. 

The mean follow-up of the entire cohort was 7 years (range: 2–16). Fourteen children reached skeletal maturity at the latest follow-up. Time to bone healing averaged 5 months (range: 1–25) and was significantly correlated with the time interval between injury and treatment (Pearson’s *R* = 0.45; *p* = 0.033) and medial displacement (Pearson’s *R* = 0.56; *p* = 0.005). Overall complications were observed in 16 patients (59%) and were significantly correlated with the initial fracture grade (*p*-value < 0.05); six were considered major (22%) and occurred in Grade 3 fractures according to Weiss, presenting lateral displacement of ≥5 mm and medial displacement of >5 mm (*p* < 0.05).

Significant correlations between baseline variables and outcomes are reported in Table 3. 

At the latest follow-up, pain and functional scores improved in 23 of 27 patients (85%) compared with the baseline. ROM increased on average by 14 degrees (*p* = 0.042). Pronation and supination were normal in all patients. MEPS increased on average from 62 points at baseline to 98 points at the latest follow-up visit (*p* < 0.0005), while DhiS improved on average from 5 to 8 points (*p* < 0.0005). Overall, outcomes were rated as excellent in 13 patients, good in 8, fair in 5 and poor in 1, according to the DhiS. Multiple stepwise linear regression showed that CDS (β-coefficient = −0.52; *p* = 0.002) and time interval between injury and treatment (β-coefficient = −0.41; *p* = 0.013) were independent predictors for MEPS at final follow-up, while only CDS was significantly associated with DhiS (β-coefficient = −0.70; *p* < 0.0005).

## 4. Discussion

Neglected fractures of the LHC in children remain a concern. Misdiagnosed or mistreated LHC fractures may occur in a relevant number of children, often requiring delayed treatment that may jeopardize the outcomes and increase the rate of complications. In our series, 22% of patients sustained major complications, in line with previous studies that reported complication rates ranging from 0 to 64%, after treatment of neglected LHC fractures [5,6,7,8,9,10,12,13,14,15,16,17,18]. This rate of complications is far higher than those following fresh fractures [1,3,4,5,11,21]. We found significant associations between the modified CDS score for complications and the MEPS and DhiS at final follow-up, confirming that poorly treated and neglected LHC fractures can be harmful for long-term elbow function and quality of life, especially in athletic children [5]. Our findings support the use of validated scores for rating complications and strengthening the requirement for widely agreed guidelines of surgical management and interpretation of results in neglected LHC fractures.

To date, we lack an agreed treatment algorithm. There is no clear definition and classification for neglected fracture of the LHC. Furthermore, no clear indication for surgical decision-making has been given [16]. In fact, while the fracture turns into nonunion, many patients become gradually asymptomatic, achieving a good range of motion and minimal disability with time. Therefore, in the past, some authors recommended against surgical fixation in children presenting more than 6 to 12 weeks after injury, preferring corrective osteotomy close to skeletal maturity when needed [5,6,14]. They worried about late surgery that could damage the blood supply of the fragment, causing AVN and decreased elbow motion. More recently, several authors have pointed out the problem of the intrinsic instability of LHC nonunion in children [12,16]. If the nonunion is left untreated, the elbow instability leads to progressive deformity, high risk of delayed cubitus valgus and tardy ulnar nerve palsy. These children are likely to require highly demanding surgery during adolescence. Therefore, there is a current consensus that LHC nonunion in children must be treated soon after presentation, although no clear guidelines for surgical decision-making have been published. We confirmed the observation that children with neglected LHC fracture, if left untreated, gradually regain full motion and function (see Appendix A). Nonetheless, this apparent recovery disappears when progressive displacement causes increasing valgus deformity, instability and tardy ulnar neuropathy [2,3,5,11,21]. These poor outcomes become obvious several years later, when the management of a long-standing nonunion requires demanding surgical procedures [18,19,20]. We found that a prolonged time interval between injury and treatment negatively impacted the treatment outcomes. Our observations are in line with some recent studies [12,15,17] supporting the recommendation for early surgical repair of established nonunion, even in asymptomatic children, to reduce unfavorable long-term sequelae. Osteosynthesis of LHC nonunion is less demanding than corrective osteotomy, having demonstrated satisfactory results in preventing deformity progression [7,8,9,10,12].

However, some unsolved questions remain. The main issue is the likelihood of identifying when a neglected LHC fracture may still heal conservatively or not within 3 months after injury. In our series, 12 children were managed within 3 months after injury. Of them, five patients with ≤5 mm displacement were treated conservatively, showing minor radiographic sequelae and overall excellent outcomes. Conversely, two children undergoing surgery within 3 months developed major complications. Furthermore, one child with >5 mm displacement refused surgery, but underwent corrective osteotomy 5.5 years later due to severe cubitus valgus. Based on our experience, we suggest caution in proposing surgery in neglected LHC fractures presenting within 3 months after injury with mild displacement (≤5 mm). Such displacement could be still acceptable, and these patients may have the chance to heal successfully by nonoperative means, if the physiological bone healing cascade is started and the fragment is stable at serial radiologic controls [5,6,14]. Surgery at this phase is crucial in major progressive displacement (>5 mm) but could be detrimental in minor displacement [6,10,14,18]. We did not observe major complications in the case of ≤5 mm lateral displacement. We hypothesize that the 2 mm threshold is too narrow for assessing the stability of neglected fractures.

Probably, at this phase, advanced imaging with computed tomography (CT) could be helpful in decision-making. We used a CT scan in doubtful cases, identifying a posteromedial bone bridge not recognizable on radiographs in some patients that healed successfully without operation (see Figure 1c). This bone bridge probably initiated close to the nutrition vessels and ensured sufficient stability for successful progression of fracture healing. Unfortunately, our study was not designed to test the accuracy, reliability and timing of CT in detecting early bone bridging. Nonetheless, other authors recommended CT scans before surgery, if a continued lack of healing is observed within 3 months [11].

The choice between ISF and ORIF remains controversial, especially in LHC nonunion (>3 months). ISF is technically less demanding and has a low risk of AVN but is jeopardized by the difficulty of restoring joint congruity to perform firm fixation and adequate curettage of fibrous tissue at the nonunion site, increasing the risk of nonunion recurrence. Conversely, ORIF usually requires extensive soft tissue dissection to adequately mobilize the LHC, which may compromise vascularity, causing AVN. Prakash et al. compared two cohorts of children undergoing ORIF or ISF for neglected LHC fracture and found that radiological union was better in the ORIF group, while the functional results and rate of complications were better in the ISF group [16]. Our study does not clarify this issue, since we did not find any superiority between these two procedures. Nevertheless, we observed that ORIF was performed in more severe cases, with wider displacement (average: 8 mm versus 4 mm) and a longer time interval between injury and treatment (average: 24 months versus 9 months), compared with ISF. These findings are consistent with some recent reports recommending ISF for minor displacement and a short time interval between injury and treatment, although no thresholds for decision-making have yet been established [13,16,17]. Based on our experience, we proposed 5 mm of lateral displacement and a 9 month time interval between injury and treatment as possible thresholds for deciding between ISF and ORIF, but further studies are needed to confirm this assumption. 

Finally, we reported three patients who underwent medial closing-wedge varus osteotomy and ATUN, without any attempt to repair the nonunion. The rationale for leaving the nonunion untreated was to prevent AVN or decrease in elbow mobility, since, in long-lasting nonunion, part of the elbow motion occurs at the level of the pseudarthrosis. Of these patients, one developed valgus recurrence with persistent ulnar neuropathy. Recently, some authors reported encouraging results from combined ISF and supracondylar dome osteotomy, which could increase the rate of successful correction, reducing the rate of complications and recurrence [19,20].

Based on our observations and on the available literature, we propose a treatment algorithm (Figure 4) that may help surgeons in decision-making regarding neglected LHC fractures:-Neglected LHC fractures presenting within 3 months after injury may be treated conservatively if the lateral displacement is <5 mm and the condylar fragment remains stable in serial radiographs. CT scans could help to identify bony bridges between fragments.-Neglected LHC fractures presenting within 3 months after injury may be treated by ISF if the lateral displacement is ≤5 mm but the condylar fragment migrates in serial radiographs, or by ORIF if the lateral displacement is >5 mm.-Neglected LHC fractures presenting at more than 3 months after injury must be treated surgically. ISF should be recommended in the case of minor displacement (≤5 mm) and early presentation (9 months on average in our cohort). ORIF should be preferred in the case of major displacement (>5 mm) and late presentation (24 months on average in our cohort). Corrective osteotomy (possibly combined with ISF and ATUN) should be recommended in case of LHC nonunion associated with severe elbow deformity.


### Limitations

Our study has limitations. Concerning the methodology, our study is a retrospective single-center case series of children undergoing different treatments for neglected LHC fracture. This is a common characteristic of almost all previous studies investigating neglected LHC fractures in children and greatly limits the strength of our recommendations. Concerning the sample size, we failed to reach the number required for detecting the potential effects of each baseline variable on the outcomes. Based on an a priori sample size calculation for multiple regression, 77 patients would be required to detect, with sufficient power (80%), at least large effect sizes (*f*^2^ = 0.35) within the correlational analyses, including all the potential predictors (20 variables). This number far exceeds the sample size of our cohort and, more generally, of all the case series currently available in the literature, which included 3 to 45 patients (16 patients per study on average). For these reasons, the correlations reported in our study must be interpreted with great caution. Finally, the lack of homogeneous groups of treatment and the insufficient length of follow-up in almost half of the cases introduced biases and prevented any conclusion concerning the effectiveness of different treatments. Prospective multicenter studies are needed to produce more generalizable and reliable treatment guidelines.

## 5. Conclusions

In conclusion, our study reports results from a cohort of patients with neglected LHC fractures, identifying potential factors that could affect the clinical and functional outcomes and influence surgical decision-making. Evidence-based algorithms of treatment and reliable thresholds to assess fracture stability are still needed to improve the management of these demanding cases.

## Figures and Tables

**Figure 1 children-08-00056-f001:**
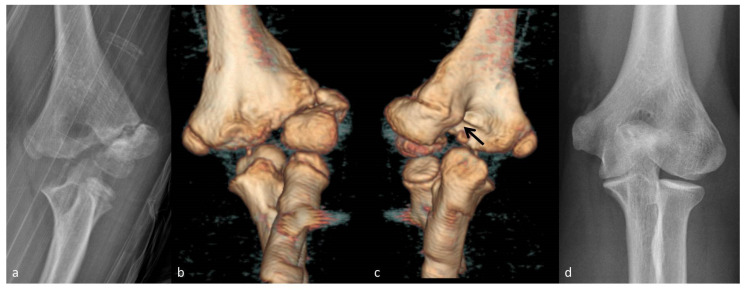
Seven-year-old boy presenting with neglected lateral humeral condyle (LHC) fracture 8 weeks after injury (Patient 1; Appendix A for further details). (**a**) Anteroposterior elbow radiographs at presentation. (**b**) Three-dimensional computed tomography (CT) reconstruction showing the anterior aspect of the neglected LHC fracture. (**c**) Three-dimensional CT reconstruction showing the posterior aspect of the neglected LHC fracture. The black arrow indicates the posteromedial bone bridge. (**d**) Follow-up radiographs 7 years after injury.

**Figure 2 children-08-00056-f002:**
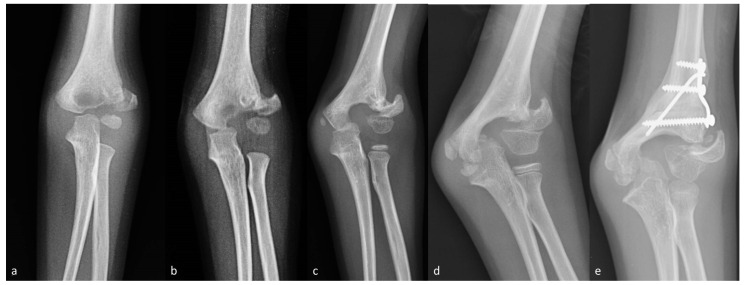
Four-year-old boy presenting with neglected LHC fracture 3 months after injury (Patient 6; see Appendix A for further details). (**a**) Anteroposterior elbow radiographs at presentation. (**b**) One year after injury. (**c**) Two years after injury. (**d**) Five years after injury. (**e**) Postoperative radiographs after CO.

**Figure 3 children-08-00056-f003:**
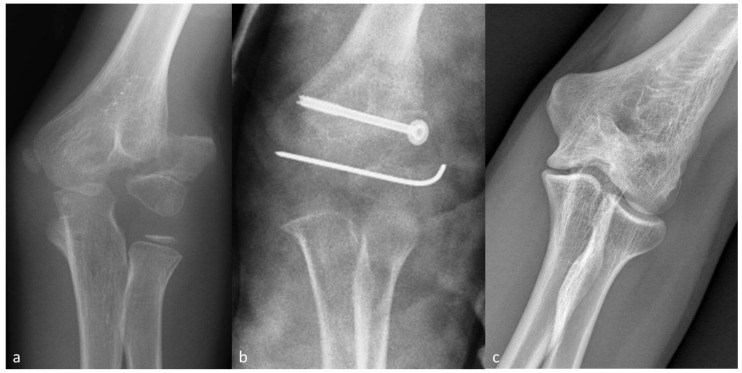
Eight-year-old boy presenting with neglected LHC fracture 7 months after injury (Patient 14; see Appendix A for further details). (**a**) Anteroposterior elbow radiographs at presentation. (**b**) Postoperative radiographs. (**c**) Follow-up radiographs 12 years after injury. Hardware was removed 15 months after surgery.

**Figure 4 children-08-00056-f004:**
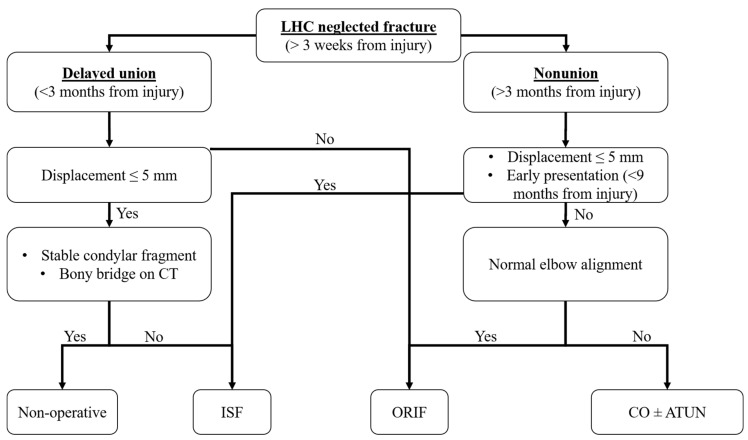
Treatment algorithm of neglected LHC fractures.

**Table 1 children-08-00056-t001:** Demographics and baseline characteristics among the different treatment groups.

Baseline Variable	Treatment				
	Conservative	ISF	ORIF	CO	Total
Patients (female/male)	6 (2/4)	7 (1/6)	11 (3/8)	3 (1/2)	27 (7/20)
Age (years)	4.8 ± 1.4 (4–7)	5.0 ± 2.2 (2–8)	5.9 ± 2.2 (3–9)	14 ± 1.0 (13–15) ***	6.3 ± 3.3 (2–15)
Time interval between injury and treatment (months)	2 ± 1 (1–3)	9 ± 16 (2–45)	24 ± 36 (1–95)	127 ± 31 (92–152) ***	27 ± 45 (1–152)
Follow-up (years)	6.8 ± 4.4 (2–13)	7.6 ± 4.1 (2–15)	8.6 ± 4.1 (2–16)	2.3 ± 0.6 (2–3)	7.3 ± 4.2 (2–16)
Weiss classification (II-III)	4-2	4-3	2-9	1-2	11-16
Song classification (III-IV-V)	5-1-0	7-0-0	3-6-2 *	1-2-0	10-9-2
Lateral displacement (mm)	4 ± 2 (3–7)	4 ± 1 (3–6)	8 ± 3 (5–12) ***	13 ± 4 (10–18)	7 ± 4 (3–18)
Medial displacement (mm)	7 ± 5 (2–16)	8 ± 3 (4–13)	10 ± 4 (4–16)	13 ± 5 (10–19)	9 ± 4 (2–19)
Carrying angle (degrees)	5 ± 4 (0–7)	−8 ± 22 (−37–10)	4 ± 17 (−30–25)	33 ± 12 (21–45) *	5 ± 20 (−37–45)
ROM (degrees)	86 ± 33 (45–140)	109 ± 35 (40–145)	98 ± 50 (20–145)	135 ± 9 (125–140)	192 ± 41 (20–145)
DhiS overall/functional (points)	5 ± 2/3 ± 2	6 ± 2/3 ± 2	5 ± 2/4 ± 2	3 ± 0/3 ± 1	5 ± 2/3 ± 2
MEPS (points)	58 ± 25 (30–95)	66 ± 17 (35–85)	60 ± 29 (10–100)	63 ± 3 (60–65)	62 ± 2 (10–100)

ISF: “in situ” fixation; ORIF: open reduction and internal fixation; CO: corrective osteotomy; ROM: range of motion; DhiS: Dhillon Score; MEPS: Mayo Elbow Performance Score; *: the difference was statistically significant with a *p*-value < 0.05; ***: the difference was statistically significant with a *p*-value < 0.0005. Bonferroni’s correction was applied for multiple comparisons.

**Table 2 children-08-00056-t002:** Complications and outcomes at the most recent follow-up among the different treatment groups.

Outcome Variable	Treatment				
	Conservative	ISF	ORIF	CO	Total
Follow-up (years)	6.8 ± 4.4 (2–13)	7.6 ± 4.1 (2–15)	8.6 ± 4.1 (2–16)	2.3 ± 0.6 (2–3)	7.3 ± 4.2 (2–16)
Time to union (months)	6 ± 4 (4–10)	3 ± 3 (1–13)	5 ± 7 (1–25)		6.3 ± 3.3 (1–25)
Overall complications	4 (67%)	2 (29%)	9 (82%)	1 (33%)	16 (59%)
Major complications (CDS > 2)	1 (17%)	1 (14%)	3 (27%)	1 (33%)	6 (22%)
Carrying angle (degrees)	13 ± 18 (0–50)	3 ± 8 (−10–15)	5 ± 16 (−20–20)	18 ± 10 (8–37)	8 ± 13 (−20–50)
ROM (degrees)	138 ± 4 (130–140)	139 ± 13 (110–145)	127 ± 25 (70–145)	143 ± 3 (140–140)	134 ± 18 (70–145)
DhiS overall/functional (points)	8 ± 1/6 ± 0.5	8 ± 1/6 ± 0.5	8 ± 1/5 ± 1	7 ± 3/5 ± 2	8 ± 2/5 ± 1
MEPS (points)	98 ± 3 (95–100)	100	98 ± 6 (80–100)	87 ± 23 (60–100)	97 ± 8 (60–100)

ISF: “in situ” fixation; ORIF: open reduction and internal fixation; CO: corrective osteotomy; CDS: Clavien–Dindo–Sink Classification System; ROM: range of motion; DhiS: Dhillon Score; MEPS: Mayo Elbow Performance Score. Bonferroni’s correction was applied for multiple comparisons.

**Table 3 children-08-00056-t003:** Correlations between baseline variables and outcomes.

Variables	Spearman’s *r*	*p*-Value
Time interval between injury and treatment/follow-up MEPS	0.43	0.026
CDS/follow-up DhiS (overall)	−0.70	<0.0005
CDS/follow-up DhiS (function)	−0.42	0.029
CDS/follow-up MEPS	−0.55	0.003
Weiss/follow-up Dhis (overall)	−0.44	0.020
Weiss/follow-up MEPS	−0.40	0.042
Song/follow-up DhiS (function)	−0.48	0.017
Baseline elbow carrying angle/follow-up elbow carrying angle	0.69	0.001
Baseline ROM/follow-up ROM	0.43	0.026
Baseline DhiS (overall)/follow-up DhiS (function)	0.48	0.012
Baseline DhiS (function)/follow-up ROM	0.44	0.022
Baseline DhiS (function)/follow-up DhiS (function)	0.44	0.023
Baseline MEPS/follow-up ROM	0.41	0.033
Baseline MEPS/follow-up DhiS (function)	0.46	0.017

Only correlations with Spearman’s *r* > 0.4 and *p*-value < 0.05 are reported. ROM: range of motion; DhiS: Dhillon Score; MEPS: Mayo Elbow Performance Score.

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
