# Peer review of "Neglected Fractures of the Lateral Humeral Condyle in Children; Which Treatment for Which Condition?"

_children, 2021, doi:10.3390/children8010056_

Round 1

Reviewer 1 Report

Thank you to the authors for submitting this manuscript and the opportunity to review it. The authors describe a retrospective case series of neglected fractures of the lateral humeral condyle in children. The topic interesting to the reader and scientific community but I am not sure how novel it is based on a number of studies describing surgical outcomes in this cohort. However, just because a study has been done before does not mean it cannot be repeated by different surgeons, authors, different institutions etc.

In its present form, I have concerns about how the information is presented. Overall, the information presented is not in a suitable format. It is very difficult to read and understand from the manuscript, especially the results and discussion sections. 

I think the authors have been over-reaching with the results and the conclusions they have drawn from the results. The associations between outcomes and variables is a very long bow to draw for this study design and this number of cases. On that note, there is also conflicting statements regarding sample size and case numbers. The authors state on multiple occasions that this is a 'large series' of neglected fractures but then state in the limitations that there was a small sample size. I do not understand this. 

Further, the way that the individual cases have been presented with the radiographs are extremely confusing and misplaced. In one section, the text in the results follows on from the figure 1 caption it seems and not the text prior to it? The cases are good to present as figures but the case descriptions should be shorter if at all presented - rather look at the outcomes as a whole for the cases and report them in that format.  

I think the manuscript should be re-written and the results presented in a more descriptive manner, rather than attempting to find associations between variables and outcomes which is not appropriate for this study design and sample size. Instead, describing the cases themselves, the surgical treatments instituted and the outcomes would be more beneficial and easier to read for the audience. However, i question whether the small number of cases and associated treatments and then outcomes, have enough weight to come up with a treatment algorithm - you would need much more cases and a prospective study design. 

See some specific comments below:

Title: I think there should be a semi colon or hyphen between children and which. Neglected fractures of the lateral humeral condyle in children; which treatment for which condition?

Line 34: 'large series' - This is not a large case series

Line 42: should ne non-displaced rather than 'undisplaced'

Line 47: "misdiagnoses may occur up to" - may occur in up to....

Lines 56-57: reword this sentence

Line 59: 'bone-grafting' is a stand alone treatment option for a neglected fracture? 

Line 62: The aim of this study WAS to describe....

Line 82: Clinical AND functional scores...

Line 97. Patient demographics not patient's

Line 102: Significant correlations ARE reported

Line 122: (n=16) AND secondary displacement....

Line 126: why 3 months? what is the significance of this? why have you said this timepoint?

Line 129: what is important flexion contracture? 

Line 138: 'accomplished' is a weird choice of word and is out of context

Lines 138-139: '...thus the study was not randomized' - I dont think you need to say this. we can already tell this by it being a retrospective study. This sentence should be reserved for the limitations sections of the discussion. 

Figure 1 Line 142: 'Seven year OLD boy'

Line 167: 'refresh' - this is weird wording for a surgical procedure. I think just delete and keep as debridement

figure 3. any information about implant removal? 

Line 181: why is this difference clinically relevant? this is not a result but this is something you have inferred from the results and is more discussion. Just state the results in the results. 

Line 195: 5+/- 6 months - this just looks weird. The SD is greater than the mean. I think just report the median and range for all outcome measures in this study. 

Line 201: pronation and supination were 'full'? what does this mean? wrong terminology. Do you mean pronation and supination were normal?

Line 205: the opening sentence of the discussion is a nothing sentence. I think it is better to restate here what the aim of the study was and what was the most meaningful finding of the study. 

Lines 230-234: this is a nothing paragraph and just sticks out, this needs to be expanded upon or incorporated into another paragraph to be anything of substance.

Line 243: again what is the relevance of 3 months post injury? 

Lines 240-241: i don't understand how these results could ever be statistically significant? what is the outcome measure? This point gets back to the overall approach by the authors to the results and what they mean. This should be more of a descriptive papers describing the outcomes for different ways of managing neglected fractures and not trying to find statistically significant results. 

Line 251: why would you use MRI for fracture assessment in these cases?

Line 268: small sample size mentioned here as a limitation but 'large series' mentioned on line 274 and in abstract on line 34

Author Response

Thank you to the authors for submitting this manuscript and the opportunity to review it. The authors describe a retrospective case series of neglected fractures of the lateral humeral condyle in children. The topic interesting to the reader and scientific community but I am not sure how novel it is based on a number of studies describing surgical outcomes in this cohort. However, just because a study has been done before does not mean it cannot be repeated by different surgeons, authors, different institutions etc.

In its present form, I have concerns about how the information is presented. Overall, the information presented is not in a suitable format. It is very difficult to read and understand from the manuscript, especially the results and discussion sections. 

I think the authors have been over-reaching with the results and the conclusions they have drawn from the results. The associations between outcomes and variables is a very long bow to draw for this study design and this number of cases. On that note, there is also conflicting statements regarding sample size and case numbers. The authors state on multiple occasions that this is a 'large series' of neglected fractures but then state in the limitations that there was a small sample size. I do not understand this. 

Further, the way that the individual cases have been presented with the radiographs are extremely confusing and misplaced. In one section, the text in the results follows on from the figure 1 caption it seems and not the text prior to it? The cases are good to present as figures but the case descriptions should be shorter if at all presented - rather look at the outcomes as a whole for the cases and report them in that format.  

I think the manuscript should be re-written and the results presented in a more descriptive manner, rather than attempting to find associations between variables and outcomes which is not appropriate for this study design and sample size. Instead, describing the cases themselves, the surgical treatments instituted and the outcomes would be more beneficial and easier to read for the audience. However, i question whether the small number of cases and associated treatments and then outcomes, have enough weight to come up with a treatment algorithm - you would need much more cases and a prospective study design. 

R: We thank the reviewer for these valuable comments. Neglected fractures of the LHC in children remain a concern. As highlighted in our manuscript, misdiagnosed or mistreated LHC fractures may occur in a relevant number of children, often requiring delayed treatment that may jeopardize the outcomes.

Despite it, to date, the treatment outcomes of neglected LHC fractures in children are underreported (approximately 400 cases described in literature) and we lack an agreed treatment algorithm. There is no clear definition and classification for neglected fracture of the LHC. Also, no clear indication for surgical decision making has been given [Prakash 2018]. In fact, while the fracture turns into nonunion, many patients become gradually asymptomatic, achieving good range of motion and minimal disability with time. Therefore, in the past, some authors recommended against surgical fixation in children presenting more than 6 to 12 weeks [Jakob 1971, Flynn 1975, Dhillon 1988], recommending corrective osteotomy close to skeletal maturity, when needed. They worried about late surgery that could damage the blood supply of the fragment, causing AVN and decreased elbow motion.  More recently, several authors have pointed out the problem of the intrinsic instability of LHC nonunion in children [Eamsobhana 2015, Prakash 2018]. If the nonunion is left untreated, the elbow instability leads to progressive deformity, high risk of delayed cubitus valgus and tardy ulnar nerve palsy, requiring demanding surgery during adolescence. Currently, there is consensus that LHC nonunion in children must be treated surgically, but no clear guidelines for surgical decision making have been done.

We acknowledge that our work does not provide further strong evidence for clear guidelines and recommendations around this topic, due to its retrospective design, and small cohort from a single institution. However, based on our experience, we believe that some concepts could be emphasized, and some key points stressed, to better approach this condition, possibly systematizing the current knowledge around this topic.

First, concerning LHC neglected fractures, a clear distinction between delayed union and nonunion must be done. Delayed union is a fracture that has slowed down, but still not stopped, the healing process, while nonunion is a fracture that will not heal spontaneously. We confirmed, in agreement with previous literature [Agarwal 2012, Eamsobhana 2015, Prakash 2018], that 3 months is a reasonable cutoff to distinguish between delayed union and nonunion. Interestingly, we found that nonoperative watchful management with prolonged immobilization is still possible within 3 months from injury if the displacement is ≤ 5 mm and no further migration of the condylar fragment is detectable on serial radiographs. In such cases, MRI or CT could reveal a bony bridge (generally not visible on plain radiographs) that is sufficient to stabilize the fracture and allow bone healing, without further problems. Therefore, we suggest caution in operating neglected fractures with minimal displacement within 3 months from injury.

Second, we reported that ISF can be more effectively attempted in case of early presentation of LHC nonunion and minor displacement (≤ 5 mm). Our findings are consistent with previous reports [Knight 2014;Park 2015; Prakash 2018], suggesting that ISF is a suitable option especially for early detected minimally displaced nonunion, reducing surgical times and risks of complications. Conversely, ORIF should be preferred in case of major displacement, and long-lasting nonunion, if no major axial deformities or ulnar nerve dysfunction have appeared in the meantime. Otherwise, corrective osteotomy and nerve transposition may improve symptoms.

In an effort to make our study more appealing for the journal readers, we re-wrote the results in a more descriptive manner, and remove a large part of the statistical analysis, according to your suggestion. We also re-wrote the discussion and added an experience-based treatment algorithm, as required by the Reviewer 3. We hope that this would be more beneficial and easier to read for the audience.

See some specific comments below:

 Title: I think there should be a semi colon or hyphen between children and which. Neglected fractures of the lateral humeral condyle in children; which treatment for which condition?

R: the title was edited according to the reviewer’s suggestion.

Line 34: 'large series' - This is not a large case series

R: thank you, we changed the manuscript accordingly.

Line 42: should ne non-displaced rather than 'undisplaced'

R: thank you, we changed the manuscript accordingly.

Line 47: "misdiagnoses may occur up to" - may occur in up to....

R: thank you, we changed the manuscript accordingly.

Lines 56-57: reword this sentence

R: thank you, we reworded the sentence according to your suggestion.

Line 59: 'bone-grafting' is a stand alone treatment option for a neglected fracture? 

R: thank you for this important comment.  As described by Flynn and colleagues, four patients with established non-union were treated by bone grafting and further stabilization with screws or pins. Only one case in this cohort of five patients was treated with bone graft alone, reporting excellent clinical results. We corrected the manuscript.     

Line 62: The aim of this study WAS to describe....

R: thank you, we changed the manuscript accordingly.

Line 82: Clinical AND functional scores...

R: thank you, we changed the manuscript accordingly.

Line 97. Patient demographics not patient's

R: thank you, we changed the manuscript accordingly.

Line 102: Significant correlations ARE reported

R: thank you, we changed the manuscript accordingly.

Line 122: (n=16) AND secondary displacement....

R: thank you, we changed the manuscript accordingly.

Line 126: why 3 months? what is the significance of this? why have you said this timepoint?

R: thank you for the question. As clarified in the introduction, “..nonunion is defined as no progression of bone healing after 12 weeks. [1, 5, 11-13]..” (line 54-55). There is no established threshold to distinguish between delayed union and nonunion in pediatric LHC fractures, but many authors report 12 weeks or 3 months irrespectively; therefore, we corrected the manuscript in a more congruous way.

Line 129: what is important flexion contracture?

R: We thank the reviewer for this question. According to Morrey [Clin Orthop Relat Res. 2005 Feb;(431):26-35], flexion contracture is defined as passive loss of extension greater than 30°. The manuscript was corrected accordingly.

Line 138: 'accomplished' is a weird choice of word and is out of context

R: thank you, we changed the manuscript accordingly.

Lines 138-139: '...thus the study was not randomized' - I dont think you need to say this. we can already tell this by it being a retrospective study. This sentence should be reserved for the limitations sections of the discussion. 

R: thank you, we changed the manuscript according to your suggestion.

Figure 1 Line 142: 'Seven year OLD boy'            

R: thank you, we changed the manuscript according to your suggestion.

Line 167: 'refresh' - this is weird wording for a surgical procedure. I think just delete and keep as debridement

R: thank you, we changed the manuscript according to your suggestion.

Figure 3. any information about implant removal? 

R: thank you, the hardwares were removed 15 months after surgery for parents’ decision since no complications occurred and patient was asymptomatic. We added a sentence in the figure 3 legend.

Line 181: why is this difference clinically relevant? this is not a result but this is something you have inferred from the results and is more discussion. Just state the results in the results.

R: thank you. We removed this sentence from the results and added it in the discussion.

Line 195: 5+/- 6 months - this just looks weird. The SD is greater than the mean. I think just report the median and range for all outcome measures in this study. 

R: thank you, we changed the manuscript.

Line 201: pronation and supination were 'full'? what does this mean? wrong terminology. Do you mean pronation and supination were normal?

R: thank you, we changed the manuscript.

Line 205: the opening sentence of the discussion is a nothing sentence. I think it is better to restate here what the aim of the study was and what was the most meaningful finding of the study.

R: thank you, we extensively revised the discussion.

Lines 230-234: this is a nothing paragraph and just sticks out, this needs to be expanded upon or incorporated into another paragraph to be anything of substance.

R: the paragraph has been removed from the revised version of the manuscript, according to your suggestion.

Line 243: again what is the relevance of 3 months post injury? 

R: thank you, 3 months (or 12 weeks) is reported by several authors as a cut-off to distinguish between delayed union and nonunion [Agarwal 2012, Eamsobhana 2015, Prakash 2018]. These authors also reported that, after 3 months, fracture anatomy is no more identifiable since the condylar fragment is progressively displaced laterally and rotated. As a result, articular cartilage faces metaphyseal parent bone. This makes union of rotated fragment almost impossible. However, the fragment is shown to retain its blood supply and it continues to grow. With time, it outgrows the parent bed and the extravasated forms dense fibrous tissue, making anatomical reduction quite difficult.

Lines 240-241: i don't understand how these results could ever be statistically significant? what is the outcome measure? This point gets back to the overall approach by the authors to the results and what they mean. This should be more of a descriptive papers describing the outcomes for different ways of managing neglected fractures and not trying to find statistically significant results. 

R: We thank the reviewer for this comment and completely revised the discussion in a more descriptive way.

Line 251: why would you use MRI for fracture assessment in these cases?

We thank the reviewer for this question. MRI could be used instead of CT to detect early bony bridge between fragments. We used low-dose CT that is more appropriate to evaluate the bone, but the use of radiations in children remains a concern. As an alternative, MRI could be used but the higher costs and the potential need of sedation in very young children should be considered. Nonetheless, we removed the MRI from the revised version of the manuscript, since a discussion around the most appropriate imaging technique overcomes the aims of our study.

Line 268: small sample size mentioned here as a limitation but 'large series' mentioned on line 274 and in abstract on line 34

R: thank you, we changed the manuscript.

Reviewer 2 Report

I had the pleasure to review the manuscript entitled: “Neglected fractures of the lateral humeral condyle in 2 children. Which treatment for which condition?".

Summary: The authors present a single-center case series of patients with neglected fractures of the lateral humeral condyle.

General Comments:

Sequelae of misdiagnosed or mistreated fracture can be dreadful to the quality of life of patients of any age. When this happens to children, the consequent deformity might affect their normal development and therefore are often subject to surgical treatment. The topic the authors describe is important, yet the author falls short in the methodology used, the sample size, and the interpretation of the data.

There are multiple problems with how this study was conducted, but the major ones are how the data was sampled (27 patients from a 15 years window), categorized (4 groups), and analyzed (multiple hypothesis testing without any alpha correction) and rigorous reporting. 

Overall, I am very aware of the hard work the authors put into the manuscript and I appreciate it.

Author Response

Sequelae of misdiagnosed or mistreated fracture can be dreadful to the quality of life of patients of any age. When this happens to children, the consequent deformity might affect their normal development and therefore are often subject to surgical treatment. The topic the authors describe is important, yet the author falls short in the methodology used, the sample size, and the interpretation of the data.

There are multiple problems with how this study was conducted, but the major ones are how the data was sampled (27 patients from a 15 years window), categorized (4 groups), and analyzed (multiple hypothesis testing without any alpha correction) and rigorous reporting. 

Overall, I am very aware of the hard work the authors put into the manuscript and I appreciate it.

R: We thank the reviewer for these valuable comments. As stated in the limitations section, we acknowledge that our study is weakened by the small number included and the retrospective way in which patients were identified. According to the reviewer’s suggestion we decided to remove a large part of the statistical analysis, since we thought that over-reaching associations between variables and outcomes is not appropriate for this study. We also re-wrote in a more descriptive manner the methods, results and discussion. Hopefully, our revised manuscript meets the expectations of you and the reviewers and be considered for publication in Children MDPI.

Reviewer 3 Report

The authors present a retrospective study on 27 pediatric patients with neglected fractures of the lateral humeral condyle. This is an important and underreported topic. Therefore, some concerns must be addressed. 

The abstract should be improved at some points. E.g., MEPS is reported in Methods but not in Results. Add the follow-up time. The correlations can be left out the abstract because of the small and diverse population.

In the Methods section, the authors should make clear who assessed the patients (lines 82-85) and the radiographs (lines 77-80); were they trained, blinded and independent? Furthermore, what do the authors mean with lateral AND medial displacement in the same case (see table 1S)? Line 138: please describe your indication for each treatment more clearly.

Reported correlations in this small and diverse series should be interpreted with great caution; this should be stressed by the authors.

Discussion; based on their experience, the findings of the study and the literature: what kind of treatment algorithm would the authors recommend to the reader?

Table 3: only the middle part is relevant.

Finally, the reviewer recommends consultation with a native speaker to improve the English text.

Author Response

The authors present a retrospective study on 27 pediatric patients with neglected fractures of the lateral humeral condyle. This is an important and underreported topic. Therefore, some concerns must be addressed. 

The abstract should be improved at some points. E.g., MEPS is reported in Methods but not in Results. Add the follow-up time. The correlations can be left out the abstract because of the small and diverse population.

R: Thank you. The abstract was corrected according to your suggestion.

In the Methods section, the authors should make clear who assessed the patients (lines 82-85) and the radiographs (lines 77-80); were they trained, blinded and independent? Furthermore, what do the authors mean with lateral AND medial displacement in the same case (see table 1S)? Line 138: please describe your indication for each treatment more clearly.

R: we thank the reviewer for these questions. Radiographs were assessed by two orthopedic surgeons (GT and DA) with more than ten-year experience in pediatric orthopedics. Clinical and functional outcomes were assessed by an orthopedic surgeon with more than twenty-year experience in pediatric orthopedics and shoulder and elbow surgery, supported by two experienced residents, after adequate training. Concerning the displacement, it was measured on radiographs of anteroposterior and internal oblique view. Medial displacement was measured from medial metaphyseal end of lateral condyle fragment to its original site of distal humerus and lateral displacement was measured from lateral metaphyseal end to its original site. The maximal values measured on anteroposterior or internal oblique view was regarded as the amount of medial and lateral displacement. [Park, J Pediatr Orthop. 2015 Jun;35(4):334-40. Finally, we added a more detailed description of each treatment and provided an experience-based treatment algorithm. The methods section was implemented accordingly.

Reported correlations in this small and diverse series should be interpreted with great caution; this should be stressed by the authors.

R: we thank the reviewer for this suggestion. We removed part of the inferential analysis from the results and from the discussion and added the reviewer’s concern in the limitations section.

Discussion; based on their experience, the findings of the study and the literature: what kind of treatment algorithm would the authors recommend to the reader?

R: we thank again the reviewer for this precious suggestion. We added an experience-based treatment algorithm accordingly.

Table 3: only the middle part is relevant.

R: Ok. We maintained only the middle part of the table in the manuscript. We reported the initial part of the table in the supplementary material, since we think that some of these correlations could corroborate our assertions concerning the natural history and prognosis of untreated LHC fractures. In particular, these relationships support the observation that that children with neglected LHC fracture, if left untreated, gradually regain full motion and function. Nonetheless, this apparent recovery disappears when progressive displacement causes increasing valgus deformity, instability and tardy ulnar neuropathy

Finally, the reviewer recommends consultation with a native speaker to improve the English text.

R: Ok. We corrected the manuscript by consulting our institutional English editing service.

Round 2

Reviewer 1 Report

Thank you for your revision. I am happy with your amendments and pleased that the paper now reads more objectively and does not make any unfounded conclusions/interpretations about your data.

I have made a few more comments below: 

Line 217: consider a different word to 'dreadful'

Line 247/248: change to '5.5 years later' or 'five and a half years later'

Author Response

Thank you for your revision. I am happy with your amendments and pleased that the paper now reads more objectively and does not make any unfounded conclusions/interpretations about your data.

R: we thank the reviewer for the valuable comments and precious suggestions that allowed to significantly improve our manuscript.

I have made a few more comments below: 

Line 217: consider a different word to 'dreadful'

R: We thank the reviewer for this suggestion we used “harmful” instead of “dreadful”.

Line 247/248: change to '5.5 years later' or 'five and a half years later'

R: thank you, we edited the manuscript accordingly.

Reviewer 2 Report

Despite the efforts of the authors, I still believe this study has major flaws that make it not meaningful for the orthopedic community.

Author Response

Despite the efforts of the authors, I still believe this study has major flaws that make it not meaningful for the orthopedic community.

We thank the reviewer for this comment; however, we disagree with the opinion that our study is not meaningful for the orthopedic community.

  • Concerning the methodology, we acknowledge that our study is a retrospective, single-center, case series of children undergoing different treatments for neglected LHC fracture. This is the major flaw of our study and we cannot change the study design. However, this is a common characteristic of almost all previous studies investigating neglected LHC fractures in children. To the best of our knowledge, all but two studies in the currently available literature are retrospective case series and there are only two published prospective studies. Prakash et al compared two cohorts of patients undergoing ORIF or ISF and found that radiological union was better in the ORIF group while functional results and rate of complications were better in the ISF group (Prakash J, J Pediatr Orthop B. 2018 Mar;27(2):134-141). Ranjan et al, compared two different hardware for fracture fixation (cannulated screws vs k-wires) and found that there was no difference in clinical outcomes, irrespective of implant used, although screw fixation was better in terms of duration of slab, improvement in carrying angle and ROM Ranjan R, Indian J Orthop. 2018 Jul-Aug;52(4):423-429). Both these prospective studies did not complete the knowledge about the management of neglected LHC fractures in children; as highlighted by the authors, there is no clear definition and classification for neglected fracture of the LHC and no clear indication for surgical decision making [Prakash 2018]. In fact, while the fracture turns into nonunion, many patients become gradually asymptomatic, achieving good range of motion and minimal disability with time. Therefore, in the past, some authors recommended against surgical fixation in children presenting more than 6 to 12 weeks [Jakob 1971, Flynn 1975, Dhillon 1988], recommending corrective osteotomy close to skeletal maturity, when needed. They worried about late surgery that could damage the blood supply of the fragment, causing AVN and decreased elbow motion. More recently, several authors have pointed out the problem of the intrinsic instability of LHC nonunion in children [Eamsobhana 2015, Prakash 2018]. If the nonunion is left untreated, the elbow instability leads to progressive deformity, high risk of delayed cubitus valgus and tardy ulnar nerve palsy, requiring demanding surgery during adolescence. Currently, there is consensus that LHC nonunion in children must be treated surgically, but no clear guidelines for surgical decision making have been done.

As pointed out before, our work does not provide further strong evidence for clear guidelines and recommendations around this topic, due to its retrospective design, and small cohort from a single institution. However, based on our experience, we believe that some concepts could be emphasized, and some key points stressed, to better approach this condition, possibly systematizing the current knowledge around this topic.

First, a clear distinction between delayed union and nonunion must be done. Delayed union is a fracture that has slowed down, but still not stopped, the healing process, while nonunion is a fracture that will not heal spontaneously. We confirmed, in agreement with previous literature [Agarwal 2012, Eamsobhana 2015, Prakash 2018], that 3 months is a reasonable cutoff to distinguish between delayed union and nonunion. Interestingly, we found that nonoperative watchful management with prolonged immobilization is still possible within 3 months from injury if the displacement is ≤ 5 mm and no further migration of the condylar fragment is detectable on serial radiographs. In such cases, MRI or CT could reveal a bony bridge (generally not visible on plain radiographs) that is sufficient to stabilize the fracture and allow bone healing, without further problems. Therefore, we suggest caution in operating neglected fractures with minimal displacement within 3 months from injury.

Second, we reported that ISF can be more effectively attempted in case of early presentation of LHC nonunion and minor displacement (≤ 5 mm). Our findings are consistent with previous reports [Knight 2014; Park 2015; Prakash 2018], suggesting that ISF is a suitable option especially for early detected minimally displaced nonunion, reducing surgical times and risks of complications. Conversely, ORIF should be preferred in case of major displacement, and long-lasting nonunion, if no major axial deformities or ulnar nerve dysfunction have appeared in the meantime. ORIF allows better reduction of the condylar fragment, reducing time to bone healing and improving clinical outcomes. Corrective osteotomy is required in long lasting nonunion with severe cubitus valgus, but it should be considered rather as a “salvage procedure” of a poorly treated LHC nonunion.

  • Concerning the sample size, the reviewer is correct. We acknowledged in the limitations that we failed to reach the sample size required for detecting potential effects of each baseline variable on the outcomes. Based on an a-priori sample size calculation for multiple regression [Soper DS, 2020. https://www.danielsoper.com/statcalc], 77 cases would be required to detect at least large effect sizes within the correlational analyses, including all the potential predictors. This number far exceeds the sample size of the case series currently available in literature. We were able to find 28 studies regarding neglected LHC fractures reporting data from overall 450 patients. These studies were generally case series of 3 to 45 patients (averagely 16 patients per study). Therefore, the sample size of our cohort (27 cases) is in line with previous studies on this topic. With the numbers available, we found some moderate correlations between four baseline variables (injury duration, fracture severity, baseline clinical and functional scores, CDS score for complications) and the clinical outcomes (MEPS and DhiS). Including only those variables that were significantly associated with the outcomes, our study reached 80% power to detect large effect size (f2 = 0.55). We updated the statistical method, the results, and the discussion, highlighting these aspects in the limitations.

  • Concerning the interpretation of data, we provided raw data (see supplementary material) and adopted a more descriptive approach, as recommended by the other reviewers. Yet, we corrected our analyses, reporting the Bonferroni correction for multiple comparisons among groups. We also conducted exploratory univariable and multivariable analyses with regression models, in order to identify potential associations between baseline variables and outcomes. Interestingly, we found that the severity of perioperative complications, as rated by the CDS system, highly influenced the clinical outcomes at the most recent follow-up. This finding supports the use of validated scores for rating complications and strengthens the requirement for widely agreed guidelines for surgical management and interpretation of results in neglected LHC fractures. Finally, we noticed that the time elapsed from injury to treatment significantly influenced the clinical outcomes. This finding is consistent with some recent studies (Knight DM, 2014; Eamsobhana P, 2015; 15. Shrestha S, 2020) and confirms the current opinion that neglected LHC fractures should be managed as soon as possible, even in asymptomatic children, in order to reduce complications and improve outcomes. However, we acknowledged that our findings must be interpreted with great caution.

Based on the reviewer’s suggestion we extensively revised the manuscript. Hopefully, the revised version addresses the reviewer’s concerns and will be acceptable for publication in Children MDPI.